# Repurposing Dimethyl Fumarate Targeting Nrf2 to Slow Down the Growth of Areas of Geographic Atrophy

**DOI:** 10.3390/ijms26136112

**Published:** 2025-06-25

**Authors:** Serge Camelo

**Affiliations:** Independent Researcher, 193 Avenue du Président Wilson, 93210 Saint-Denis, France; sergecamelo2@hotmail.com; Tel.: +33-0650825448

**Keywords:** age-related macular degeneration, geographic atrophy, oxidative stress, Nrf2, dimethyl fumarate

## Abstract

Recently, marketing authorizations were granted by the Federal Drug Administration (FDA) for pegcetacoplan and avacincaptad pegol, which inhibit C3 and C5 complement components, respectively. These two drugs were demonstrated to slow down the growth of atrophic areas in the retina. These authorizations represent a huge breakthrough for patients suffering from geographic atrophy (GA), the late stage of the dry form of Age-related Macular Degeneration (AMD). Until then, no treatment was available to treat this blinding disease. However, these two new compounds inhibiting the complement system are still not available for patients outside of the United States, and they are not devoid of drawbacks, including a poor effect on vision improvement, an increased risk of occurrence of the neovascular form of AMD and the burden of patients receiving recurrent intravitreal injections. Thus, the important medical need posed by GA remains incompletely answered, and new therapeutic options with alternative modes of action are still required. Oxidative stress and inflammation are two major potential targets to limit the progression of atrophic retinal lesions. Dimethyl fumarate, dimethyl itaconate and other activators of the transcription factor nuclear factor erythroid 2-related factor 2 (Nrf2) display antioxidants and immunomodulatory properties that have shown evidence of efficacy in in vitro and in vivo models of dry AMD. Tecfidera^®^, whose active principle is dimethyl fumarate, is already commercialized for the treatment of autoimmune diseases such as multiple sclerosis and psoriasis. The aim of this review is to present the rationale and the design of the clinical trial we initiated to test the effectiveness and safety of repurposing Tecfidera^®^, which could represent a new therapeutic alternative in patients with the dry form of AMD.

## 1. Introduction

Age-related Macular Degeneration (AMD) is the leading cause of irreversible blindness in the industrialized world [1]. Approximately half of the patients with the late stages of AMD suffer from the dry form of the disease also called geographic atrophy (GA). GA is characterized by the progressive extension of atrophic lesions in the retina, linked with the loss of retinal pigmented epithelial cells (RPEs) and of photoreceptors that are essential for the visual cycle to occur. Millions of patients worldwide suffering from GA are at risk of becoming blind in the short term, and this represents a major personal and societal health problem. Until very recently, there was no therapy to cure this blinding disease or to slow the progression of retinal atrophy areas leading to vison loss. In the last couple of years, however, based on the Phase 2 Filly and the two phase 3 trials, Derby and Oaks on one hand and the Gather I and II phase 3 studies on the other, a C3 complement inhibitor (pegcetacoplan) [2] and a C5 complement inhibitor (avacincaptad pegol) [3] have shown a significant capacity to reduce the rate of progression of atrophic retinal lesions and have been granted marketing authorization by the Federal Drug Administration (FDA) in the United States [4,5,6,7,8,9]. By contrast, these two drugs have not been approved by the European Medicine Agency (EMA) in the European Union because of their important limitations. Indeed, both pegcetacoplan and avacincaptad pegol require repeated intravitreal injections every 1-2 months, have a limited efficacy in terms of visual improvement for patients and display an increased incidence of side effects such as ischemic optic neuropathy, occlusive retinal vasculitis and new onset of neovascular AMD [6,9,10,11,12,13]. Therefore, developing alternative therapeutic strategies to slow GA progression which preserve visual function with a favorable safety profile still appears necessary. These potential new drugs should also ideally be administrable orally and have a limited cost to limit the patient’s burden. One potential avenue to reach this goal is to repurpose existing drugs already commercialized with an acceptable safety profile and a relevant mode of action.

GA is a complex multifactorial disease whose main risk factor is aging [14]. GA initiation and evolution is also influenced by genetic polymorphisms in the factor H of the Complement (CfH) [15,16,17]. This discovery at the beginning of the 21st century was at the source of the clinical development of anti-complement compounds. Environmental factors such as smoking are also important due to inducing oxidative stress, which is involved in the disease initiation and evolution [18,19,20]. Oxidative stress influences AMD by modifying RPE cells physiology through mitochondrial and autophagy dysfunction [21,22,23] and by inducing an inflammatory response detrimental to the retina [24]. Indeed, the important role in AMD pathophysiology of inflammation mediated by cells of the innate immune system (mostly macrophages) has been recognized for more than a decade now [25,26,27]. Even more recently, effector T cells mediating adaptative immunity were also observed near GA lesions and have been suspected to participate in the progression of the retinal atrophic areas [28,29,30,31].

Therefore, based on this long but not exhaustive list of risk factors associated with dry AMD, repurposing therapeutic approaches targeting oxidative stress and inflammation and able to modulate the adaptative immune response could be particularly beneficial for patients risking losing vision due to GA [32]. Here, we present evidence that the transcription factor nuclear factor erythroid 2-related factor 2 (Nrf2) is a major therapeutic target potentially controlling the occurrence and evolution of GA symptoms and that Tecfidera™, administrable per OS, and whose active principle dimethyl fumarate (DMF), which possess antioxidant and immunomodulatory properties via induction of Nrf2 nuclear translocation, appears well suited to be repurposed in patients with the late stage of the dry form of AMD. Based on this observation, this review presents the rationale and the design of the clinical trial Long term Analysis of Dimethyl Fumarate to slow the growth of Areas of Geographic Atrophy (LADIGAGA) (NCT04292080), aiming to test the effectiveness and safety of repurposing Tecfidera™, which could represent a new therapeutic alternative in patients with the dry form of AMD.

## 2. Methods

We performed a targeted review following the PRISMA guidelines [33] (Figure 1) by searching publicly available data from the scientific literature on the natural history of GA as well as the basic science of oxidative stress, inflammation, adaptative immunity and Nrf2 activators in AMD and the mode of action of Tecfidera™ (Biogen Netherlands B.V. Prins Mauritslaan 13, 1171 LP Badhoevedorp, The Netherlands). Relevant studies published and available on PubMed up to the 27th of December 2024 were searched for. Using Boolean operators (e.g., AND, OR), the applied search terms included combinations of the following key words: “ocular”, “retina”, “macula”, “macular”, “age-related macular degeneration”, “AMD”, “Geographic Atrophy (GA)”, “(anti)-inflammatory”, “(anti)oxidant”, “NRF2”, “Tecfidera™”, “dimethyl fumarate (DMF)”, “itaconate”, “dimethyl itaconate (DMI)”, “macrophages”, “lymphocytes”, “interleukin-(IL)-17”, “complement”, “pegcetacoplan” and “avacincaptad”. To minimize the risk of omitting relevant studies, the reference lists of all eligible papers were also manually checked. Only publications in the English language were included.

## 3. Results

### 3.1. The Direct and Indirect Effects of Oxidative Stress in GA

The three main risk factors associated with AMD, aging, blue light exposure and cigarette smoking, are known to induce an imbalance in the redox system by increasing the excessive production of Reactive Oxygen Species (ROS) in parallel to reduced antioxidant responses [34]. These elements combined reinforce each other and lead to what could be depicted as a vicious circle, generally called; oxidative stress that is directly toxic for RPE cells and photoreceptors, and the two main cell populations degenerating during GA [35,36,37,38]. Oxidative stress induced by aging, blue light exposure and smoking is also involved in mitochondrial and mitophagy dysfunction [24,34,39,40,41,42]. Alterations of the proper functioning of mitochondria in RPE cells which produce ATP are detrimental for photoreceptors’ tips degradation and regeneration, the visual cycle process and, in general, the maintenance of the homeostasis of the retina [22]. Uncontrolled oxidative stress enhances the production of ROS by the mitochondria themselves, further fueling the imbalance of the redox system and alterations in the retina [22].

A more indirect role may also be played by oxidative stress during AMD by initiating inflammation and an abnormal “autoimmune-like” response directed against the retina [29,43]. Indeed, oxidative stress can induce the production of inflammatory cytokines such as IL-1, IL-6 and TNF-α by RPE cells in vitro and in vivo in animal models of dry AMD [44,45,46,47]. Secretion of chemokines including IL-8, CCL2 and CCL5 by RPE cells was also detected in response to oxidative stress in vitro [44,46]. In addition, oxidative stress may induce the adjunction of carboxyethyl pyrrole (CEP), malondialdehyde (MDA), 4-hydroxynonenal (4-HNE), carboxymethyl lysine (CML) and pentosidine to proteins [29,40,48]. These adjuncts could be recognized as neo-epitopes susceptible to alerting the immune system towards retinal proteins [29,43]. Accordingly, newly recruited T lymphocytes and macrophages were able to kill RPE cells and photoreceptors of mice immunized with CEP-adducted proteins [49]. In animal models of AMD following blue light illumination and aging, macrophages and T lymphocytes were observed at the level of the subretinal space [26,50,51]. In humans, increased levels of CD56-positive T cells were detected in the plasma of patients with the dry form of AMD [52], and accumulation of macrophages and T lymphocytes producing IL-17 are observed in humans at the level of retinal and RPE atrophic areas [30]. Altogether, oxidative stress induced by genetic and environmental risk factors linked to AMD appears to be at the source and plays a central role in several molecular mechanisms that are toxic to the retina.

### 3.2. Nrf2, a Potential Important Molecular Target for the Treatment of Patients with GA

Based on the preceding observations, restoration of the redox balance or limiting the toxic effects of oxidative stress could prove beneficial for patients with GA. In natura, antioxidant enzymes, necessary to keep the physiological balance and cope with excessive oxidative stress, are expressed under the control of the transcription factor Nrf2 [34,43,53,54,55,56]. In the basal state, i.e., when the intracellular level of oxidation is “normal”, the protein Kelch-like ECH-associated protein-1 (Keap1) sequesters inside the cytosol and then the Keap1/Cul3/ RBX1/E3 ubiquitin ligase complex polyubiquitinates the Nrf2 protein, leading to its degradation by the proteasome [34,56]. When excessive oxidation occurs, i.e., during oxidative stress, reactive oxidative species (ROS) interact with the cysteine 151 (Cys151) of Keap1, releasing Nrf2, which then translocates into the nucleus and attaches to the antioxidant response element (ARE) responsible for the transcription of phase II antioxidant enzymes [54]. These enzymes include Heme-oxygenase-1 (HO-1), NADPH dehydrogenase (NQO1), super-oxide dismutase 2 (SOD2), catalase, thio-redoxin 1 (Trx1), gluta-redoxin 1 (Grx1) and the γ-Glutamyl-Cyteine Ligase (GCL), catalytic (GLCc) and modifier (GCLm) subunits. Accordingly, in vitro, in a photoreceptor cell line, when submitted to oxidative stress induced following blue light illumination, the expression of Nrf2 and associated antioxidative responses genes are elevated and were proven to be neuroprotective [57].

In parallel, it has been shown that Nrf2 nuclear translocation antagonizes the activation of the inflammatory cascade through multiple mechanisms. Indeed, there is an inverse relationship between the activation of the phase II antioxidant response by Nrf2 and the production of inflammatory cytokines. For instance, Nrf2 indirectly inhibits the secretion of IL-1β through reduction of ROS formation that is necessary for activation of the nucleotide oligomerization domain, leucine-rich repeat- and pyrin domain-containing protein 3 (NLRP3) inflammasome [58]. In addition, Nrf2 directly competes with the transcription factor NF-κB (nuclear factor-κB) for liaison with transcriptional cofactors such as CBP (CREB-binding protein)–p300 [59]. Nrf2 also impairs the transcription of interleukin (IL)-6 and IL-1β through allosteric inhibition of the recruitment of the RNA polymerase II on the promoter sequences of these inflammatory cytokines [59,60]. Thus, regulation of oxidative stress and inflammation are ultimately linked through the action of Nrf2. Moreover, Nrf2 is one of the transcription factors controlling mitochondrial biogenesis that is important to respond to high energy demand, especially in stressed conditions, and whose dysregulation is associated with AMD pathology [21,22,61].

In agreement with the important role of Nrf2 in retinal homeostasis in vivo, multiple alterations of the retina, including photoreceptor and RPE cells loss and Bruch’s membrane thickening, are observed in old Nrf2^-/-^ mice. Moreover, Nrf2 deficiency in mice is associated with lipofuscin and complement accumulation, increased expression of inflammatory cytokines and recruitment of macrophages and T cells producing IL-17 in the sub-retinal space and spontaneous neo-vascularization at the level of the choroid and into the retina [62,63,64,65]. Similarly, in double-knockout mice lacking the Nrf2 and the CXCR5 genes, Huang and his colleges observed spontaneous accumulation of complement proteins and of subretinal deposits and photoreceptors degeneration recapitulating some structural characteristics of dry AMD [66]. These observations in in vivo knockout animal models suggest that Nrf2 impairs the occurrence of clinical signs like those of dry AMD leading to GA in humans, and control the inflammatory reaction as well, and are confirmed by immunohistochemistry studies of human-eye sections from patients with AMD showing that Nrf2 expression is reduced in RPE cells above drusen compared to its expression in RPE in healthy areas of the same eyes [63].

Altogether, these observations suggest that reduced expression of Nrf2 is associated with the occurrence of the first clinical signs of AMD in humans. Therefore, it has been suggested that limitation of the deleterious effects of pathological oxidative stress, inflammation and mitochondrial disfunction through enhancement of Nrf2 activity would be beneficial for patients with GA [34,67,68,69].

### 3.3. Compounds Activating Nrf2 Translocation Are Beneficial Against Retinal Degeneration

Results obtained in multiple experimental models of retinal degeneration in vitro and in vivo are in agreement with this hypothesis. Several compounds that can induce the expression of the phase II antioxidant enzymes through Nrf2 translocation protect RPE cells against various oxidative stress insults [34,69,70]. Amongst these compounds, quercetin, with antioxidant properties, has been shown in vitro to protect RPE cells from apoptosis and reduce inflammation induced by cigarette smoke extracts by stimulating Nrf2 nuclear translocation [71,72]. In normal rats in vivo, photo-oxidative damage induced with LEDs was diminished with a combination of quercetagetin and lutein/zeaxanthin that modulates the Nrf2 pathway [73]. The carotenoids lutein and zeaxanthin, which are naturally present in the macula of the retina, exert a beneficial protective antioxidant effect by absorption of UV light but also through nuclear translocation of Nrf2 stimulating the expression of antioxidant enzymes [74,75,76]. In addition, recently, it has been reported that the oral supplements AREDS/AREDS2 including lutein/zeaxanthin, which were known for many years to reduce slightly the incidence of choroidal neovascularization, also slow down the progression of GA towards the fovea of patients with dry AMD [77]. Furthermore, it has been demonstrated that other nutritional supplements, such as compound K, a metabolite of ginseng, well known for its antioxidant capacity, also preserve the ARPE-19 cell line in vitro against oxidative stress induced by H_2_O_2_ exposure, through expression of the antioxidant enzyme complex regulated by Nrf2 [78].

Itaconate (or itaconic acid) (Figure 2a) is a metabolite of the tricarboxylic acid (TCA) cycle following transformation of *cis*-aconitate by the cis-aconitate decarboxylase (ACOD-1), also called immune responsive gene 1 (IRG1), in the mitochondrial matrix when the TCA cycle is bypassed (reviewed in [79]). Itaconate and its derivatives dimethyl-itaconate (DMI) (Figure 2b) and 4-Octyl itaconate (4-OI) (Figure 2c) activate Nrf2 by alkylation of multiple cysteines residues on Keap1 [79,80]. Therefore, itaconate and its derivatives display antioxidative and anti-inflammatory properties linked with their capacity to induce the nuclear translocation of Nrf2 but also through protein modification regulation independent of keap1-Nrf2, such as inactivation of the NRLP3 inflammasome and inhibition of cytokines secretions mediated by regulation of the NF-κB-STAT-1 and JAK1-STAT6 pathways [79]. This anti-inflammatory and antioxidative effects of itaconate and its derivatives have been observed in multiple inflammatory, neurodegenerative, autoimmune and infectious diseases models [79,81,82] which are beyond the scope of the present review but also in cell types more relevant to AMD. For instance, itaconate and its derivative modulate inflammation and oxidative stress in vitro in immune cells: bone marrow derived dendritic cells, the RAW264.7 macrophages cell line, microglia and the Jurkat T cell line [83,84]. In addition, 4-OI reduced the expression of the inflammatory cytokines IL-6, IL-8 and MCP-1 and the expression of malondialdehyde and of ROS in a Nrf2-dependent manner following induction of oxidative stress by Angiotensin-II in human primary RPE cells [85]. To our knowledge, these effects of itaconate and its derivatives have not been tested in proper models of AMD; however, importantly, the in vitro results reported above are comforted by the observation in vivo that both 4-OI and DMI inhibit proinflammatory cytokines production, Th-17 T cells differentiation and cytokines expression and reduced disease severity in experimental autoimmune uveitis through Nrf2 activation [86,87].

Another specific activator of the Nrf2 translocation/antioxidant responsive elements (ARE) pathway, the synthetic triterpenoid derivative 1-[2-cyano-3-,12-dioxooleana-1,9(11)-dien-28-oyl]imidazole (CDDO-Im), also leads to the synthesis of phase II antioxidative enzymes and reduces the accumulation of complement and inflammatory cells recruitment in the subretinal space in non-genetically modified aging animals [63,64]. Similarly, treating mice with an adenovirus vector secreting in the retina TatNrf2mer, a cell-penetrating peptide targeting the Nrf2 signaling pathway, was also protective against photoreceptor loss measured by OCT, preserved the visual functions recorded via electro-retinogram and diminished the inflammatory response following a challenge with sodium iodate (NaIO_3_) [88]. In summary, all these examples are only a small sample of reported observations showing the effect of a myriad of antioxidant molecules acting through Nrf2 and that tend to confirm that activators of the Nrf2 pathway induce the production of phase II antioxidative enzymes [89,90,91], protect retinae from destruction and limit inflammation and the occurrence of symptoms similar to those observed during dry AMD in humans.

### 3.4. Repurposing Tecfidera™, Whose DMF, Its Active Principle, Is a NRF2 Activator, for the Treatment of GA

Dimethyl fumarate (DMF), also known by its chemical name, dimethyl (E) butenedioate (C_6_H_8_O_4_), is represented in Figure 2e. DMF is a methyl ester of fumaric acid (fumarate) (Figure 2d). DMF is a precursor of monomethyl fumarate (MMF) (Figure 2f), which has been shown to reduce the relapse rates and retard disability progression in patients with MS through activation of the nuclear translocation of Nrf2 [54,92]. DMF exhibits similar biological functions to DMI; however, when compared in immune cells, DMF displayed superior anti-inflammatory properties [83]. Moreover, it was demonstrated in vitro that DMF significantly reduced production of proinflammatory mediators in classically activated microglia, through an Nrf2-independent pathway [93,94]. Furthermore, DMF attenuated the neurotoxicity of activated microglia and rescued mitochondrial respiratory deficits in primary cortical neurons in vitro [93].

In AMD models, pre-treatment for 24 h with DMF in vitro protected RPE cells from oxidative stress induced using tert-butylhydroperoxide (tBH) in vitro [95]. Still, in vitro, in primary RPE cell lines obtained from AMD patients, DMF activates the synthesis and functions of glutathione (GSH) and GCL, which belongs to the phase II antioxidative enzymes [69,95]. In vivo, in normal mice subjected to blue light illumination, a model of retinal injury relevant for AMD, MMF and its precursor DMF, through activation of Nrf2, protected retinal integrity, and MMF reduced photoreceptors’ apoptosis and restored ERG amplitudes [96,97,98]. In these mice challenged by blue light, it was also shown that, in agreement with the known anti-inflammatory effects induced following Nrf2 nuclear translocation, MMF limited NLRP3, IL-1-α and TNF-α expression, and both MMF and DMF limited the recruitment/activation of macrophages/microglia in the subretinal space [96,97]. DMF is the active principle of Tecfidera™, commercialized as an oral drug and used to treat several autoimmune diseases such as relapsing Multiple Sclerosis (MS) [99]. MS, which affects the central nervous system, leads to progressive disability and shares several risk factors with dry AMD [100]. Increase in IL-4 expressing CD4^+^ T cells and decreases in IFN-γ and IL-17-expressing CD4^+^ T cells were observed in DMF-treated MS patients [101,102]. DMF also increases the inhibitory effectiveness of regulatory T cells (Tregs) by enhancing the responsiveness of effector T cells to Tregs [103]. Since IL-17 producing T cells are present within atrophic lesions and in the subretinal space of eye sections obtained from GA patients [30], this suggests that DMF may also be beneficial to treat patients suffering from the dry form of AMD by inhibiting their activation [104].

Altogether, all the in vitro and in vivo evidence strongly confirms the therapeutic potential of using Tecfidera™ containing DMF to protect RPE cells and retinal cells in, or around, atrophic areas to slow (or even perhaps block) the enlargement of degenerated patches in the retina of patients with the dry form of AMD [47,67,68,69]. The mode of action and potential beneficial effects in this indication of compounds activating the Nrf2 pathway and of DMF specifically are summarized in Table 1 and Figure 3, respectively.

### 3.5. Description of a Clinical Trial Evaluating the Efficacy of Tecfidera™ Repurposed for the Treatment of Patients with GA

Repurposing DMF for the treatment of ocular diseases has been proposed by other authors previously [105], but, despite the medical need still represented by the dry form of AMD, it has never been tested before [105]. In 2022, we started the recruitment of patients with GA resulting from the dry form of AMD in a phase II clinical trial testing the safety and efficacy of Tecfidera^®^. This study, Long term Analysis of Dimethyl Fumarate to slow the growth of Areas of Geographic Atrophy (LADIGAGA), is sponsored by “Assistance Publique Hôpitaux de Paris” (APHP) and funded by a grant from “Programme Hospitaller de Recherche Clinique (PHRC)—2017 (French Ministry of Health)” and is registered on the http://clinicaltrials.gov website (NCT04292080); accessed on 3 July 2024.

The primary objective of this trial is to compare the efficacy to reduce and/or slow photoreceptors and RPE degeneration and the associated vision loss in 30 patients with GA treated twice daily with Tecfidera^®^ (oral DMF; 120 mg the 1st week, 240 mg for 51 weeks thereafter) versus 30 patients receiving standard of care. The primary endpoint at month 12 is the rate of change in GA area based on masked, digital grading as measured on Fundus Auto-fluorescence (FAF) imaging using a confocal scanning ophthalmoscope by a central reading center compared to value at baseline (Day 0). The comparison between the two groups will be made after square root transformation using a Student test.

Changes from baseline in the following secondary criteria will be evaluated at 3, 6, 12 and 24 Months:Area of GA determined 1/ by FAF and 2/ by Fundus Photography.Macular GA determined 1/ by FAF and 2/ by Fundus Photography.Total Drusen Area (determined by FAF).Best Corrected visual Acuity (BCVA) (absolute values).Contrast Sensitivity determined using the Pelli–Robson Chart.Number of scotomatous points (determined by micro-perimetry).Mean retinal sensitivity (determined by micro-perimetry).Macular choroidal and retinal thickness.The percentage of participants with Improved, Stabilized, or worsened BCVA determined using the ETDRS chart.Number of scotomatous points and mean retinal sensitivity (determined by micro-perimetry).Changes in the NIHVFQ25 questionnaire.Changes in blood immune cell populations’ percentage, plasma CRP concentration, plasma SOD activity levels, MDA levels and ROS levels, and plasma cytokines concentration will be determined at 6, 12 and 24 months compared to baseline at Day 0.

The various tests that will be performed during the trial are summarized in Figure 4: Schematic representation of the chronology of the study.

We are inclined to believe that Tecfidera™ could be beneficial for patients with the dry form of AMD due to its proven antioxidant and anti-inflammatory properties, because its safety profile and tolerability is well-known and because of its immediate availability as an oral formulation commercialized for the treatment of MS and of psoriasis.

Nevertheless, we are aware of the limitations of our study design. For instance, one of the difficulties of repurposing a drug given systematically for an ocular indication is to determine the dosage required to obtain a clinically optimal exposure of the eye structures. Several authors have suggested that a topical or intraocular mode of administration of DMF could avoid some of the known adverse effects of Tecfidera^®^ occurring primarily at the level of the gastrointestinal tract, and, by reducing the dosage needed to obtain a biological effect, this local route of administration would increase the drug efficacy [104]. In our study, however, Tecfidera™ is administered orally and following the treatment regimen used in other indications for which it is commercialized, firstly to avoid the necessity to perform an equivalence studies and based also on the fact that it has been demonstrated that DMF, the active principle of Tecfidera™, is able to reach the retina by crossing the blood–retinal barrier [106].

Another potential concern regarding repurposing a commercially available drug in a new indication is to make sure that its mode of action is well adapted to the new disease and that it does not lead to potential safety issues. In our case of repurposing Tecfidera™, we and others have reported that enhanced expression and nuclear translocation of Nrf2 has multiple potential beneficial effects in AMD, as in several other diseases where oxidative stress and inflammation are at play, such as periodontitis, as an example [107]. On the contrary, however, it is well known that Nrf2 activation has been associated with deleterious effects in human cancers by increasing cancer cells survival, metastasis and drug resistance [108]. These processes are partly linked with the central role of Nrf2 in the regulation of ROS generation and the induction of an antioxidative stress response, the inhibition of apoptosis and ferroptosis and the regulation of autophagy. Discussing the reasons behind these opposite effects of Nrf2 in these different indications is beyond the scope of this review, but it could be hypothesized that the hyperproliferative status of cancer cells versus the non-proliferative status of the quiescent RPE and photoreceptors that are the major cell populations in the human retina degenerating during AMD could explain this discrepancy. Nevertheless, we took this potential cancer-promoting risk of Nrf2 activation by DMF into account during the design of our trial, and, to avoid any risk of cancer aggravation, participants with a history of malignancy that would compromise the 2-year study survival or with history of cancer diagnosed within the past five years that could be worsened by immunosuppression have been specifically excluded.

Similarly, to mitigate the potential risk of adverse events (AEs) for patients with AMD, especially in the treated group, we follow all contraindications and recommendations described in annex 1 of the Tecfidera™ product information available on the European Medicine Agency website [109].

For instance, the most reported adverse reactions (incidence ≥10%) for patients treated with Tecfidera™ are flushing and gastrointestinal events (i.e., diarrhea, nausea, abdominal pain, abdominal pain upper). To reduce the occurrence of this AE, patients with severe active gastrointestinal disease are excluded, and we also remind patients and the clinical team that Tecfidera™ is better tolerated when taken with food and, moreover, that two healthy-volunteer studies have shown that the occurrence and severity of these AEs could be reduced following a short course of treatment with 75 mg non-enteric coated acetylsalicylic acid.

Lymphopenia is also a common AE (≥1/100 to <1/10) reported in patients administered with Tecfidera™. Following heath agencies’ recommendations, patients with lymphocytes count below normal laboratory values at inclusion cannot participate in our study. More broadly, any screening laboratory value (hematology, serum chemistry or urinalysis) three times above normal values or that in the opinion of the investigator is clinically significant and not suitable for study participation leads to the non-enrolment of patients in our study. For patients included in the trial, and only for those randomized in the group treated with Tecfidera™, complete blood counts, including lymphocyte counts, are performed every 3 months (see Figure 4).

Progressive Multifocal Leukoencephalopathy (PML) is an opportunistic infection caused by the John Cunningham virus, which may be fatal or result in severe disability. Even though no global increase in incidence of infections has been reported following Tecfidera™ treatment, close monitoring of lymphocyte counts is important in relation to the knowledge that PML cases have occurred with Tecfidera™ and other products containing fumarates in the setting of moderate to severe prolonged lymphopenia [110]. PML incidence is low and estimated to be 0.02 per 1000 patients (1/50,000) with relapsing remitting MS treated with dimethyl fumarate [102,110]. To further reduce the risk of the development of PML in our study in GA patients randomized in the Tecfidera™ group, a baseline MRI is required (within 3 months) or must be performed within one month to confirm inclusion. In the absence of sign or symptom suggestive of PML, the patient’s inclusion is confirmed and treatment with Tecfidera™ can be initiated (see Figure 4). By contrast, if the MRI reveals any sign or symptom suggestive of PML, the patient is excluded from the study without receiving any administration of the study treatment.

## 4. Conclusions and Perspective

Despite the challenges ahead, its limited sample size and its unblinded nature, we hope that this trial will confirm the good safety profile of Tecfidera™ in this aging population suffering from GA and help to identify beneficial visual effects for patients. Hopefully, this will increase awareness of the potential of repurposing DMF, and Nrf2 activators in general, to treat the dry form of AMD. If the results of this study are positive or promising enough, we hope this avenue will get more attention from the scientific and medical community and will lead to further preclinical and clinical studies so that this oral drug or other treatments with a similar mode of action could potentially benefit patients with GA in the future.

## Figures and Tables

**Figure 1 ijms-26-06112-f001:**
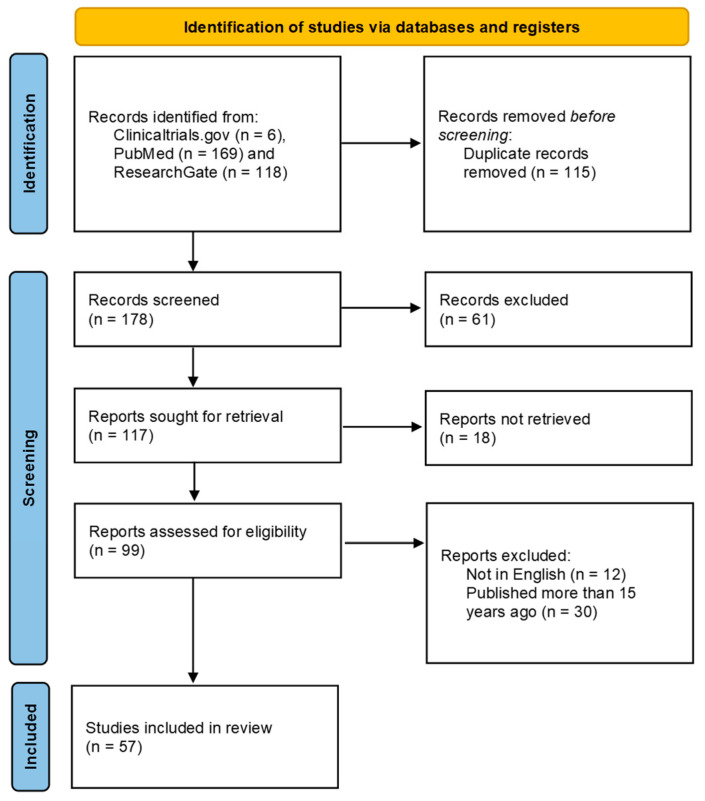
PRISMA 2020 flow diagram for new systematic reviews, which included searches of databases and registers only.

**Figure 2 ijms-26-06112-f002:**
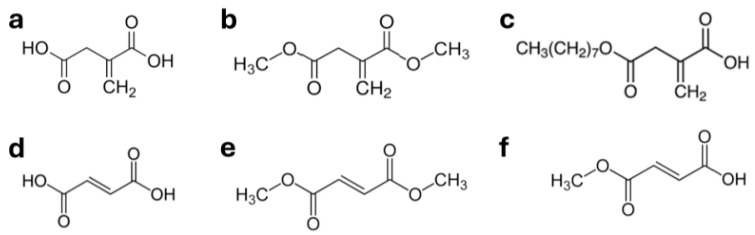
Chemical structures of some Nrf2 activators: (**a**) itaconic acid (itaconate), (**b**) dimethyl itaconate (DMI), (**c**) 4-Octyl itaconate, (**d**) fumaric acid (fumarate), (**e**) dimethyl fumarate (DMF) and (**f**) monomethyl fumarate (MMF).

**Figure 3 ijms-26-06112-f003:**
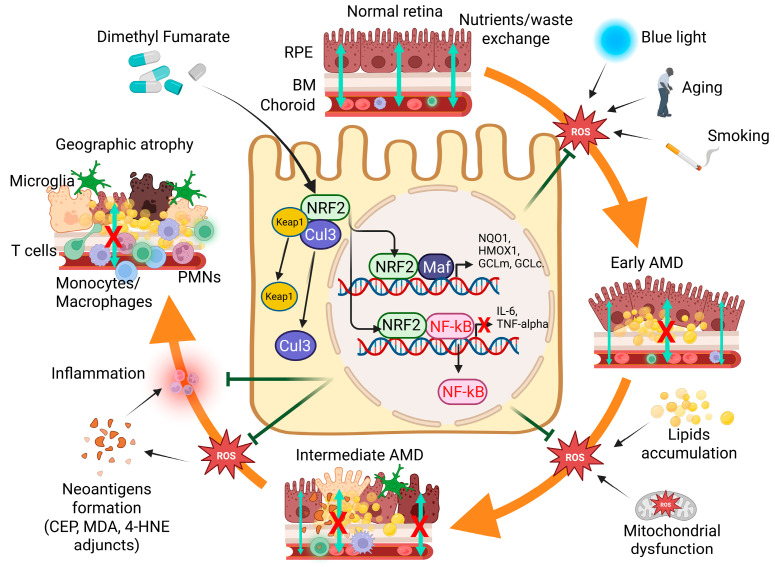
Role of oxidative stress and potential effects of Nrf2 activation by dimethyl fumarate in AMD clinical evolution based on the Beckmann classification system. Orange arrows represent the steps of the natural evolution of AMD. Green arrows represent potential effects induced following Nrf2 activation by Tecfidera™ susceptible to interfere with this evolution. AMD: age-related macular degeneration; GA: geographic atrophy; ROS: reactive oxygen species; monocytes/macrophages/microglia (M); T lymphocytes (T); Neutrophils (PMNs); nuclear factor erythroid 2-related factor 2 (Nrf2); Kelch-like ECH-associated protein 1 (KEAP1), Cullin 3 (CUL3), Heme-oxygenase-1 (HO-1); NADPH dehydrogenase (NQO1); thio-redoxin 1 (Trx1); gluta-redoxin 1 (Grx1) and the γ-Glutamyl-Cyteine Ligase (GCL), catalytic (GLCc) and modifier (GCLm) subunits; carboxyethyl pyrrole (CEP); malondialdehyde (MDA); 4-hydroxynonenal (4-HNE). Created in https://BioRender.com.

**Figure 4 ijms-26-06112-f004:**
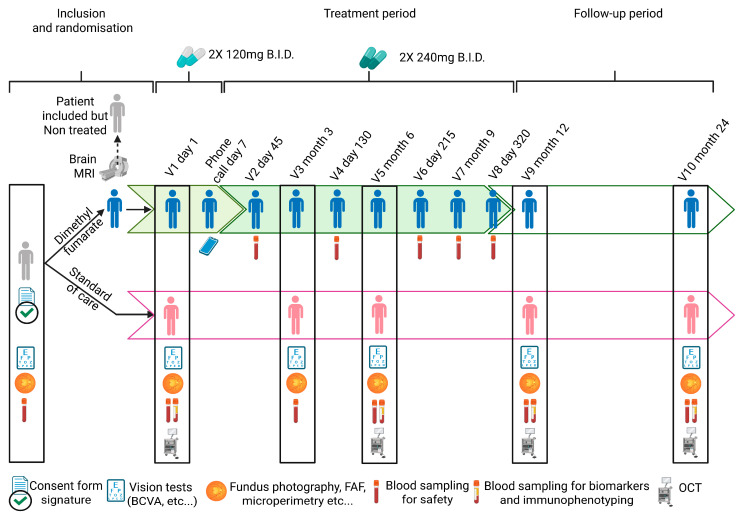
Schematic representation of the chronology of the tests that will be performed during the trial. Created in https://BioRender.com.

**Table 1 ijms-26-06112-t001:** Summary of compounds activating Nrf2 with a potential beneficial effect in dry AMD.

Compound	Reported Effects Relevant for AMD Physiopathology	Reference(s)
Quercetin	Protects RPE cells from apoptosis and reduces inflammation induced by cigarette smoke extracts in vitro.	[71,72]
Quercetagetin + lutein/zeaxanthin combination	Reduces photo-oxidative damage induced with LEDs in vivo in rats.	[73]
Lutein and zeaxanthin	Reduce oxidative stress by inducing the expression of antioxidant enzymes in vivo.	[74,75,76]
AREDS/AREDS2 oral supplements containing lutein/zeaxanthin	Reduce the speed of progression of GA towards the macula in vivo in humans.	[77]
Compound K (a metabolite of ginseng)	Preserves the ARPE-19 cell line against oxidative stress induced by H2O2 exposure. Induces the expression of the antioxidant enzyme complex in vitro.	[78]
Itaconate	Reduces inflammation and oxidative stress in dendritic cells, macrophages/microglia and the T cell line in vitro.	[83,84]
Dimethyl-itaconate (DMI)	Reduces inflammation and oxidative stress in dendritic cells, macrophages/microglia and the T cell line in vitro.	[83,84]
4-Octyl itaconate (4-OI)	Reduces inflammation and oxidative stress in dendritic cells, macrophages/microglia and the T cell line in vitro. Reduces the expression of IL-6, IL-8, and MCP-1 and of malondialdehyde and of ROS induced by Angiotensin-II in human primary RPE cells in vitro.	[83,84,85]
1-[2-cyano-3-,12-dioxooleana-1,9(11) -dien-28-oyl] imidazole (CDDO-Im)	Induces the synthesis of phase II antioxidative enzymes. Reduces the accumulation of complement and inflammatory cells recruitment in the subretinal space during aging in vivo.	[63,64]
TatNrf2mer (an AAV delivered cell penetrating peptide)	Reduces photoreceptor loss, preserves visual functions and diminishes inflammation induced by sodium iodate (NaIO3) in vivo.	[88]
17beta-Estradiol	Suppress light induced retinal degeneration in rats in vivo.	[89]
Monomethyl fumarate (MMF)	Protects retinal integrity, reduces photoreceptors apoptosis, reduces retinal inflammation and restores ERG amplitudes in vivo in a blue-light illumination model.	[96,97,98]
Dimethyl fumarate (DMF) (precursor of MMF)	Reduces cytokines production by microglia, reduces microglia’s neurotoxicity and protects cortical neurons in vitro.	[93,94]
	Induces antioxidant enzymes production and protects RPE cells from oxidative stress induced by tert-butylhydroperoxide (tBH) in vitro.	[69,95]

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
