# Peer review of "Repurposing Dimethyl Fumarate Targeting Nrf2 to Slow Down the Growth of Areas of Geographic Atrophy"

_ijms, 2025, doi:10.3390/ijms26136112_

Round 1

Reviewer 1 Report

Comments and Suggestions for Authors

In this manuscript author present the rationale of a clinical trial aimend to test the effectiveness and safety of repurposing Tecfidera® in patients with the dry form of AMD.

Although the topic is interesting, the manuscript needs several improvements. See my comments below.

Introduction: Author must state what is the aim of the manuscript

Line 11: "... two drugs..." which ones, add

2. Methods: Since author used PRISMA, a flow diagram illustrating the article selection process is necessary

Lines 133-134: nuclear factor 133 erythroid 2-related factor 2 (Nrf2) has already been full length written. Abbreviate with just Nrf2

Lines 134-136: is not KEAP1 that polyubiquitinates Nrf2 but the KEAP1/CUL3/RBX1 E3-ubiquitin ligase complex to which Nrf2 is bound

Lines 134-156: The mustifaceted role of Nrf2/keap1 signaling deserves to be highlighted since this pathway is involved in the onset and progression of several cancerous and non-cancerous diseases (see PMID: 39456522, PMID: 39267682). 

3.3. Compounds activating Nrf2 translocation are beneficial against retinal degeneration: A table summarizing the main results of the studies discussed in this section would be useful

An accurate revision of syntax is necessary

Reviewer 2 Report

Comments and Suggestions for Authors

In this manuscript, the authors summarized the effectiveness and safety of Tecfidera as an activator of the transcription factor Nrf2 in the treatment of GA. This short review is very comprehensive and logical, providing substantial information about the drug tests.

Minor Comments:

  1. In section 3.1, the authors mentioned that AMD is associated with aging, blue light exposure, and smoking. Regarding Tecfidera, whose active component DMF activates Nrf2 for the treatment of GA, how do aging, blue light exposure, or smoking affect the treatment process? If there are any published studies, could the authors discuss them?

  2. Please confirm the font size is consistent in lines 158–160.

Round 2

Reviewer 1 Report

Comments and Suggestions for Authors

the manuscript has been significantly improved and can be accepted in the present form